# Dynamic Functional Network Connectivity Changes Associated with fMRI Neurofeedback of Right Premotor Cortex

**DOI:** 10.3390/brainsci11050582

**Published:** 2021-04-30

**Authors:** Zhiying Long, Zhaoxi Guo, Zhitao Guo, Hang Zhang, Li Yao

**Affiliations:** 1The State Key Laboratory of Cognitive Neuroscience and Learning & IDG/McGovern Institute for Brain Research, Beijing Normal University, Beijing 100875, China; 201621430057@mail.bnu.edu.cn (Z.G.); gzt2019@mail.bnu.edu.cn (Z.G.); yaoli@bnu.edu.cn (L.Y.); 2Institute of Psychological Sciences, College of Education, Hangzhou Normal University, Hangzhou 310031, China; kervinhang@hznu.edu.cn; 3The School of Information Science and Technology, Beijing Normal University, Beijing 100875, China

**Keywords:** real-time fMRI, neurofeedback, dynamic functional network connectivity, functional connectivity state, motor execution

## Abstract

Neurofeedback of real-time functional magnetic resonance imaging (rtfMRI) can enable people to self-regulate motor-related brain regions and lead to alteration of motor performance and functional connectivity (FC) underlying motor execution tasks. Numerous studies suggest that FCs dynamically fluctuate over time. However, little is known about the impact of neurofeedback training of the motor-related region on the dynamic characteristics of FC underlying motor execution tasks. This study aims to investigate the mechanism of self-regulation of the right premotor area (PMA) on the underlying dynamic functional network connectivity (DFNC) of motor execution (ME) tasks and reveal the relationship between DFNC, training effect, and motor performance. The results indicate that the experimental group spent less time on state 2, with overall weak connections, and more time on state 6, having strong positive connections between motor-related networks than the control group after the training. For the experimental group’s state 2, the mean dwell time after the training showed negative correlation with the tapping frequency and the amount of upregulation of PMA. The findings show that rtfMRI neurofeedback can change the temporal properties of DFNC, and the DFNC changes in state with overall weak connections were associated with the training effect and the improvement in motor performance.

## 1. Introduction

Real-time functional magnetic resonance imaging (rtfMRI) technology has been widely applied to train individuals to actively control their brain activities by using real-time neurofeedback [1,2] in recent years. Many rtfMRI studies have demonstrated that individuals can self-regulate brain activity in a target brain region or connectivity in a target network to improve the corresponding cognitive ability and behavior performances [3,4,5,6,7,8]. Moreover, rtfMRI neurofeedback was applied to train a number of clinical populations, such as those with chronic stroke [9], schizophrenia [10] and Parkinson’s disease [11,12], to help the rehabilitation of patients.

Previous rtfMRI studies have shown that subjects can self-regulate brain activity in a target motor-related region or connectivity between motor-related regions. Some studies reported that rtfMRI was able to train subjects to voluntarily modify activities in the somatomotor cortex, left M1, and left premotor cortex without measuring motor performance [13,14,15,16]. Some rtfMRI studies that measured motor performance demonstrated that self-regulation of the supplementary motor area or right premotor areas (PMA) can induce an improvement in motor performance [3,11,12]. Moreover, the neurofeedback of connectivity between the left primary motor cortex and the left lateral parietal cortex can induce the aimed direction of change in connectivity and also a differential change in cognitive performance [17]. The above studies suggest that self-regulation of the motor-related area or connectivity can change motor performance.

Besides modulation of motor performance, neurofeedback training of activities in motor-related regions or connectivity between motor-related regions can also modify functional connectivity (FC) between brain regions/networks. The neurofeedback training of regulating the right PMA induced an increase in FC between the target region PMA and the other motor-related regions [3,18]. Moreover, the altered FC was related to motor performance [3]. In Horovitz’s study (2010), the regulation of the motor cortex not only induced correlations between the visual and motor areas but also increased basal ganglia involvement and bilateral motor cortex connectivity during neurofeedback training [19]. Self-regulation of the supplementary motor area (SMA) can be achieved by Huntington’s disease patients and leaded to an increase in FC between the SMA and left putamen, with beneficial effects in motor behavior [20]. rtfMRI neurofeedback training of the connectivity between the left lateral parietal and left primary motor areas modified FC between the motor/visuospatial network and the default model network [17,21].

Although previous rtfMRI studies revealed changes in FC underlying motor tasks after self-regulating the activities of motor-related areas, the FC used in these studies was considered in a “static” sense. The static FC neglected the dynamic characteristics of fMRI data during the whole scanning process [22]. Moreover, an increasing amount of evidence suggests that FCs of fMRI data fluctuate over time [23,24]. The temporal fluctuations of FC at different time scales are crucial to understanding the ultimate neural underpinnings of cognitive processing. In recent years, dynamic FC analysis has been applied to fMRI data widely and has revealed reoccurring patterns of dynamic FC, called dynamic states [22,25,26]. However, few studies have investigated the relationship between dynamic FC fluctuations and rtfMRI neurofeedback training so far. In particular, the effect of neurofeedback of motor-related areas on the dynamic FC of motor execution tasks is unclear. Investigating the reorganization of dynamic FC underlying motor tasks after neurofeedback of motor-related areas is essential for us to further understand the neural mechanism of neurofeedback training and reveal the relationship between changes in dynamic FC and neurofeedback training.

Our previous studies investigated changes in static FC with the neurofeedback of the right dorsal PMA (dPMA) [3]. In this study, we were particularly interested in exploring the mechanism of self-regulation of the right dPMA with rtfMRI neurofeedback on the underlying dynamic functional network connectivity (DFNC) of motor execution (ME) tasks. For this purpose, we first conducted independent component analysis (ICA) on the fMRI data of motor execution (ME) tasks before and after neurofeedback training to extract task-related networks. We focused on interactions within and between task-related networks that were positively or negatively related with the ME task. Then, time-varying FC within and between task-related networks was estimated using a series of sliding windows. We applied K-means clustering to FC matrices of all the windows to identify connectivity states that reoccurred over time and were reproducible across subjects.

In this study, we aimed to test whether right dPMA self-regulation modulated DFNC underlying the motor execution (ME) task and revealed the relation among the neurofeedback training effect, DFNC, and motor performance. Previous studies revealed that dynamic FCs changed with development [27], aging [28], long-term training [29], and disease [30,31]. Given that DFNC was plastic, we hypothesized that the neurofeedback of the right dPMA would change the DFNC of some FC states of the ME task. Furthermore, several previous studies have shown that the temporal properties of DFNC are correlated with some behavior performance, such as personality traits, clinical score, social abilities, and attention scores [32,33,34,35]. Moreover, some rtfMRI studies found that FC changes were correlated with behavioral improvements after the neurofeedback training [3,36]. Given the relationship between DFNC/FC and behavior measures, we hypothesized that the DFNC changing underlying ME tasks is related to motor behavior improvement after self-regulation of the right dPMA.

## 2. Materials and Methods

The fMRI data in this study were the same as in our previous study [3].

### 2.1. Participants

Fifteen right hand-dominant participants (6 males and 9 females, mean age: 22 years, SD: 1.6 years) in the experimental group and 13 right hand-dominant participants (8 males and 5 females, mean age: 23, SD: 1.7 years) in the control group participated in the experiment. All subjects gave written informed consent, which was set by the MRI Center of Beijing Normal University. The experiment was approved by the Institutional Review Board (IRB) of the State Key Laboratory of Cognitive Neuroscience and Learning in Beijing Normal University (approval number ICBIR_B_0011_006).

### 2.2. Imaging Parameters

The fMRI scanning was performed on a 3T Siemens scanner at the MRI Center of Beijing Normal University. A single-shot T2*-weighted echo EPI sequence (TR = 2000 ms, TE = 40 ms, flip angle = 90°, FOV = 240 × 240 mm^2^, matrix = 64 × 64, slice thickness = 4 mm, inter-slice gap = 0.8 mm, slices = 32) was used in the experiment. The EPI sequence was the same for the whole fMRI experimental procedure.

### 2.3. Experimental Procedure

The whole experimental procedure included pre-scan practice, a pre-training scan, rtfMRI neurofeedback training, a post-training scan, and a questionnaire interview. Figure 1 shows the diagram of the whole fMRI experimental procedure.

#### 2.3.1. Pre-Scan Practice

All subjects practiced tapping their right index finger according to a metronome at 4 Hz to adapt the frequency used in the experiment. The subjects were instructed that their four fingers from index finger to little finger represented one, two, three, and four separately. Then, they tapped 1-2-3-4 and 4-2-3-1-3-4-2 at 4 Hz for a 30-s period and imagined tapping the 4-2-3-1-3-4-2 at 4 Hz for a 30-s period without actual movement. The experimental and control groups received the same pre-scan training.

#### 2.3.2. Pre- and Post-Training Scan

Both the pre- and post-training scan included one ME run and one motor imagery (MI) run. Five resting 30-s blocks alternated with four 30-s task blocks in each 270 s run. When the word “PUSH” was displayed on the screen, the subjects needed to tap or imagine the sequence of 4-2-3-1-3-4-2 at 4 Hz with their right hand. When subjects saw “Rest” on the screen, they needed to remain awake and relax.

#### 2.3.3. Neurofeedback Training

For each subject, one target ROI and one background ROI were manually defined in TBV software based on the activation map during the pre-training ME run. The target ROI was located in the right dorsal PMA and the background ROI that excluded the motor-related areas covered almost all voxels around the axial slice (MNI: z = 10). The neurofeedback value that was calculated based on Equation (1) was presented to the subjects.
Neurofeedback value = (ROI_t-task_ − ROI_t-rest_) − (ROI_b-task_ − ROI_b-rest_)(1)

ROI_t-task_ and ROI_b-task_ in Equation (1) are the mean fMRI signals of the target ROI and background ROI during the task block. ROI_t-rest_ and ROI_b-rest_ are the mean fMRI signals of the two ROIs during the preceding rest block.

There were four neurofeedback training runs. Each run lasted 7.5 min. Eight 30-s rest blocks were alternated with 7 30-s task blocks in each run. During the neurofeedback runs, all participants were asked to increase the curve height of the neurofeedback as much as they could by imagining the series of 4-2-3-1-3-4-2 without actual finger movement. The strategy instructions that were provided to all subjects included varying the speed, strength, and the method of finger tapping and so on during motor imagery. During the rest blocks, the subjects were asked to be relaxed without any movement. The only difference between the experimental group and the control group was that the control group received sham neurofeedback that was selected randomly from the experimental group.

#### 2.3.4. Questionnaire Interview

After the post-training scan, all the subjects were required to finish a questionnaire about their performance. In addition, all participants were asked questions about their states in the experiment.

### 2.4. Offline Data Analysis

Because this study mainly investigated the alterations of DFNC of the ME tasks that were induced by neurofeedback training, the fMRI data of the MI tasks were excluded from the following analysis in this study. Moreover, 12 experimental subjects and 12 control subjects were used in the following analysis because 3 subjects in the experimental group and 1 subject in the control group reported finger movements during the movement imagery based on our previous study [3]. An overview of the framework of DFNC processing is presented in Figure 2.

#### 2.4.1. Preprocessing

The whole preprocessing procedure was carried out in the SPM8 package (http://www.fil.ion.ucl.ac.uk/spm/software/spm8, released April 2009). For each subject, functional images of the pre-training ME run, the four neurofeedback training runs, and the post-training ME run were corrected for slice timing and head movement. Then, the images were spatially normalized into a standard EPI template, resliced to 3 × 3 × 4 mm^3^ voxels, and spatially smoothed with an 8 mm full-width half maximum (FHWM) Gaussian kernel.

#### 2.4.2. Group ICA

Group spatial ICA [37] in GIFT 3.0 version (http://mialab.mrn.org/software/gift/, released 21 May 2013) was applied to the pre-training and post-training fMRI data of the ME tasks for all subjects. Prior to the group ICA analysis, a two-step principal components analysis (PCA) was used to reduce data dimensionality to 100 based on the previous study [25]. The Infomax ICA algorithm [38] was repeated 100 times using the ICASSO algorithm to obtain stable and reliable components [39]. GICA1 back-reconstruction was performed to estimate the time course and spatial map of each component of each subject.

After the back-reconstruction, the correlation coefficient between the mean time course of each component over all subjects and the reference function was calculated. The reference function was derived from the convolution of the task paradigm with the hemodynamic response function (HRF). The components with an absolute correlation coefficient larger than 0.5 were selected as the task-related components. Thirty-five selected components underwent further analysis. One sample *t*-test was applied to each selected component of all the subjects to identify the activation map of each component. The components whose spatial activation map contained ventricles, vascular, white matter area, and cerebrospinal fluid artifacts were further excluded from the following analysis. The final 29 task-related components were retained for the DFNC analysis. Based on the functions of Brodmann areas (BA), anatomical properties, and network division of the previous DFNC study [25], the 29 selected components were divided into four network groups: motor-related network (MN), visual network (VN), cognitive control network (CCN), and default mode network (DMN).

#### 2.4.3. DFNC Analysis

The whole process of DFNC analysis was performed using the sliding windows approach of the “temporal DFNC” toolbox in GIFT 3.0 version [25,40]. The time courses of all the selected components were triple detrended (linear, quadratic, cubic), despiked, and lowpass filtered (0.15 Hz) [41]. For the pre-training/post-training run of each subject, the Pearson correlation between the time courses of each pair of components during a series of sliding windows was calculated. Gaussian windows were used (δ = 3 TR), and the step of the sliding windows was set to 1 TR, resulting in 121 windows. Because 30–60 s could be a reasonable choice of window lengths [42], we set the length of each window to 30 s, which was the length of each block. For each sliding window, the correlations of all selected components consisted of an FC matrix with c × c size, where c is the number of the selected components. To reduce possible noise from a limited number of data in each window, L1 regularization was applied to the inverse covariance matrix to promote sparsity. The regularization parameter was optimized for each subject independently by evaluating the log-likelihood of windowed covariance matrices from the same subject in a cross-validation framework [25,40,41]. Finally, there were 121 FC matrices for the pre-training/post-training run of each subject.

#### 2.4.4. Clustering Analysis

The FC matrices of the pre-training and post-training runs for all subjects were clustered by K-means to identify reoccurring FC patterns [25]. L1 distance (Manhattan distance) was used in K-means because it may be a more effective similarity measurement for high dimensional data [43]. The number of clusters was set to 7 according to the Akaike information criterion method [44] (see Appendix A). The FC matrix exemplars with local maxima were selected as centroids first, and K-means was repeated 150 times to obtain stable cluster results subsequently. The median of each cluster was extracted as the “FC state.”

#### 2.4.5. Temporal Property Analysis of DFNC

For each state, the mean dwell time and fraction of time [39] were calculated in this study. The mean dwell time of a dynamic FC state was measured as the average number of consecutive windows corresponding to the state in a run, whereas the fraction of time of a state was measured as the proportion of windows corresponding to the state during a single run. The mean dwell time denotes how long a participant remains in a particular state and fraction of time denotes the probability of occurrence of a state. Two-way repeated measure analyses of variance (ANOVA) that used training as the within-subject factor (pre-training and post-training) and group as the between-subject factor (experimental and control group) were conducted for the mean dwell time and fraction time of each state. All two-way repeated measure ANOVAs were carried out in SPSS 20.0 (https://www.ibm.com/analytics/data-science/predictive-analytics/spss-statistical-software, released August 2011). Among all dynamic FC states, the FC states that showed significant main effects and interaction effects were selected as the target dynamic FC states. The target FC states underwent further analysis. The effect sizes of the ANOVAs were calculated by the online WebPower [45].

#### 2.4.6. Subnetwork Connectivity Analysis of the Target FC States

For each target state, the FC matrix was divided into 10 connectivity patterns that were FC within the MN (M-M), within the VN (V-V), within the CCN (C-C), within the DMN (D-D), between the MN and VN (M-V), between the MN and CCN (M-C), between the MN and DMN (M-D), between the VN and CCN (V-C), between the VN and DMN (V-D), and between the CCN and DMN (C-D). We called the 10 connectivity patterns the 10 subnetworks in this study. For each target state, the strength of each subnetwork was computed by averaging all the Fisher Z scores of the correlation coefficients across all the connections in the subnetwork. For each subnetwork of each group, two-way repeated measure ANOVAs were performed to investigate the training and group effect. The effect sizes of the ANOVAs were calculated by the online WebPower.

#### 2.4.7. Relationship between the Behavioral Performance and DFNC

For each target FC state, the mean dwell time, the fraction of time, and the strengths of the 10 subnetworks were used to examine the relationship between the behavioral performance (tapping frequency) and DFNC. The correlation between the behavioral performance and the mean dwell time/fraction of time/subnetwork strength was calculated for each target state of each group’s pre-training and post-training run.

#### 2.4.8. Relationship between the Neurofeedback Training Effect and DFNC/Motor Behavior

Offline GLM analysis was applied to each subject’s preprocessed data of the pre-training ME run and the four neurofeedback training runs using the SPM8 package. For each subject’s pre-training ME run, the target ROI was defined as a 6 mm-radius sphere around the maxima of the right dPMA based on the t-contrast map that compared the ME task with the baseline. The percent signal change (PSC) of each voxel for each training run was computed by dividing the beta value corresponding to the task regressor by the beta of the constant term. The mean PSC of the target ROI was calculated by averaging PSC across voxels within the ROI.

For each target FC state, the mean dwell time, the fraction of time, and the subnetwork strength were used to investigate the relationship between the neurofeedback training effect and DFNC. The correlation between the PSC of each training run and the mean dwell time/fraction of time/subnetwork strength was calculated for each target state of each group’s pre-training and post-training run. Moreover, the correlation between the PSC of each training run and the motor behavior was calculated for each target state of each group’s pre-training and post-training run.

#### 2.4.9. Network-Based Statistic (NBS) Analysis

For each target FC state, the NBS method that was demonstrated to yield substantially greater statistical power than generic methods for controlling the family-wise error [46] was used to identify the network connections that showed significant group/training differences. For the target state of each subject’s pre-training/post-training run, the FC matrices of the sliding windows that belonged to the target state were transformed into Fisher Z scores, and the mean FC matrix was obtained by averaging across the corresponding sliding windows.

For each group, paired *t*-tests were applied to each pairwise connectivity of the mean FC matrices of the pre-training and post-training runs to identify the training differences. For the pre-training/post-training run, a two-sample *t*-test was applied to each pairwise connectivity of the mean FC matrices of the experimental and control groups to identify the group differences. The test statistic for each pairwise connectivity was thresholded (*t* > 3) to construct a set of suprathreshold links. Based on the set of suprathreshold links, the connected components were detected using a breadth first search algorithm from the GRETNA toolbox (https://www.nitrc.org/projects/gretna, released 19 November 2017). The number of links (size) for each connected component was calculated.

A permutation test was carried out for each *t*-test to ascribe a *p*-value that was controlled for the FWE to each connected component based on its size. Each permutation test generated 1000 random permutations. For each permutation of two-sample *t*-test, all subjects were randomly reallocated into two groups, and the two-sample *t*-test was recalculated. For each permutation of the paired *t*-test, the pre-training and post-training runs for all subjects in each group were randomly assigned to either the pre-training or post-training run, and the paired *t*-test was recalculated. The same threshold (*t* > 3) was used to generate a set of suprathreshold links, and the maximal component size of each connected component was stored. The maximal component size of all the permutations was then ranked in ascending order. For each observed component of size k, the *p*-value is estimated by calculating the ratio M/1000, where M is the number of permutations whose maximal component size is greater than k. The components with *p* < 0.05 were selected as the suprathreshold links that showed significant group/training differences. For each group, the correlation between the alteration of behavioral performance and the alteration of the suprathreshold links that survived the NBS test was also calculated.

## 3. Results

### 3.1. Temporal Property Analysis of DFNC

The spatial activation maps of the 29 task-related components that belonged to the MN, VN, CCN, and DMN are presented in Appendix A. Moreover, Figure 3 shows the median patterns of seven FC states. The seven states are differentiated by connectivities between components. State 1 showed strong positive correlations between MN/VN/CCN/DMN components and strong negative correlations between MN and VN/CCN/DMN components. State 2 showed weak positive correlations between MN/VN/CCN/DMN components and slight correlations that were close to zero between MN and VN/CCN/DMN components. State 3 had high positive correlations between MN components and weak positive correlations for all the other connections. State 4 displayed weak positive FCs between MN/VN/CCN/DMN components and negative FCs between MN and VN/CCN/DMN components. The modular structure of state 5 was not as clear as the other states. State 6 displayed high positive FCs between MN components, weak positive FCs between VN/CCN/DMN components, and weak negative FCs between MN and VN/CCN/DMN components. State 7 showed high positive FCs between most MN components, weak positive FCs between VN/CCN/DMN components, and negative FCs between MN and VN/CCN/DMN components.

Among the seven FC states, the mean dwell times of state 2 and state 6 were the highest two. Figure 4 presents the comparison of the mean dwell time and the fraction of time for the states that showed significant group/training differences. The repeated measure ANOVA tests revealed a significant main effect of group for both the mean dwell time (F_group_(1,22) = 10.156, *p* = 0.004 < 0.05, effect size = 0.6686) and the fraction of time (F_group_(1,22) = 13.092, *p* = 0.002 < 0.05, effect size = 0.9867) of FC state 2. Tests of simple effect revealed that the experimental group showed significantly lower mean dwell time (F(1,22) = 10.12, *p* = 0.004 < 0.05, effect size = 1.2988) and fraction of time (F(1,22) = 21.09, *p* < 0.001, effect size = 1.8748) than the control group for state 2 in the post-training ME run. Moreover, there was no significant training main effect, group main effect, or interaction effect for the mean dwell time of state 6. However, a significant group main effect (F_group_(1,22) = 7.416, *p* = 0.012 < 0.05, effect size = 0.6433) was observed in the fraction of time of state 6. The simple effect analysis revealed that the fraction of time of the experimental group was significantly higher than that of the control group for state 6 in the post-training run (F(1,22) = 6.86, *p* = 0.016 < 0.05, effect size = 1.0691). FC states 2 and 6 were selected as the target states for further analysis.

### 3.2. Subnetwork Connectivity Analysis of the Target FC States

Figure 5 shows the strength of the subnetworks that showed significant group/training differences for the target states (state 2 and state 6). For the strength of subnetwork D-D within DMN in state 2, the group main effect was significant (F_group_(1,22) = 6.358, *p* = 0.019 < 0.05, effect size = 0.6202). Tests of simple effect showed that the D-D strength of the control group was significantly higher than that of the experimental group after the neurofeedback training (F(1,22) = 8.36, *p* = 0.008 < 0.05, effect size = 1.172). Moreover, the strength of subnetwork C-D between the CCN and DMN of state 6 showed a significant main group effect (F_group_(1,22) = 5.962, *p* = 0.023 < 0.05, effect size = 0.5726). Tests of simple effect showed that the C-D strength of the experimental group was significantly higher than that of the control group after the neurofeedback training (F(1,22) = 5.50, *p* = 0.028 < 0.05, effect size = 0.9604).

### 3.3. Relationship between Behavioral Performance and DFNC

Figure 6a,b display the relationship that has a significant correlation between the behavioral performance and the mean dwell time/fraction time of the target FC states. The experimental group showed significant negative correlation between the tapping frequency and the mean dwell time (*r* = −0.5560, *p* = 0.0302 < 0.05) or the fraction of time (*r* = −0.5562, *p* = 0.0309 < 0.05) for state 2 of the post-training ME run. No significant correlations were observed in state 6 of the experimental group or in the two target states of the control group. Moreover, the subnetwork strengths of state 2 and state 6 were not significantly correlated with the motor performance of the two groups.

### 3.4. Relationship between the Neurofeedback Training Effect and DFNC/Behavioral Performance

Figure 7 shows the relationship that had a significant correlation between the PSC of the training runs and the mean dwell time/fraction time of the target FC states. For the experimental group, the mean dwell time of state 2 in the post-training ME run showed significant negative correlation with the PSC of run C (*r* = −0.7038, *p* = 0.0106 < 0.05) and run D (*r* = −0.5836, *p* = 0.0451 < 0.05). Moreover, the fraction time of state 2 in the post-training ME run showed significant negative correlation with the PSC of run C (*r* = −0.6246, *p* = 0.0299 < 0.05). No significant correlations were observed in state 6 of the experimental group or in the two target states of the control group. Moreover, the neurofeedback training effect did not show significant correlation with motor performance.

Figure 8 shows the relationship that had a significant correlation between the PSC of the training runs and the subnetwork strength of the target FC states. For the experimental group, the D-D strength of state 2 was significantly negatively correlated with the PSC of run C (*r* = −0.6107, *p* = 0.0349 < 0.05) and run D (*r* = −0.5949, *p* = 0.0413 < 0.05). For the control group, the V-C strength of state 6 was significantly positively correlated with the PSC of run C (*r* = 0.6175, *p* = 0.0324 < 0.05) and run D (*r* = 0.6841, *p* = 0.0141 < 0.05).

### 3.5. NBS Analysis

By using the NBS method, connections showing significant differences between the pre-training and post-training were found in state 2 of the experimental group (*p* < 0.05) and in state 6 of the control group (*p* < 0.05) (see Figure 9a,b). For state 2 of the experimental group, real-time neurofeedback training resulted in a significant increase in two MN-DMN connections and two MN-VN connections. Moreover, one MN-VN connection, one VN-DMN connection, two MN-CCN connections, and one MN-MN connection were significantly decreased after neurofeedback training. For state 6 of the control group, there were 13 connections that showed significant differences between the pre-training and post-training runs. Among the 13 connections, 11 connections between the component of the bilateral middle occipital gyrus and the 11 components of MN showed a significant increase after neurofeedback training. Furthermore, one connection between DMN and MN was significantly enhanced, whereas another connection between two DMN components was significantly decreased after neurofeedback training. The details of the statistical results in state 2 and state 6 can be found in Table 1 and Table 2, respectively.

Among all the connections, there were only two connections whose alteration showed significant correlation with the alteration of the behavioral performance after the neurofeedback training for state 2 of the experiment group. Figure 9c,d show the relationship between the alteration of the two connections and the alteration of the behavioral performance after the neurofeedback training. Moreover, the mean strength of the two connections during the pre-training and post-training runs for both groups are shown in Figure 9e,f. For the connection between the bilateral precentral gyrus component and the hippocampus component (connection 1), the increase in the tapping frequency was significantly negatively correlated with the increase in the connection strength after the neurofeedback training (*r* = −0.677, *p* = 0.015) (see Figure 9c). Moreover, the mean strength of connection 1 was significantly decreased in the experimental group after the neurofeedback training (see Figure 9e). For the connection between the bilateral postcentral gyrus component and the middle occipital gyrus component (connection 2), the increase in the tapping frequency showed a significantly positive correlation with the increase in the connection strength after the neurofeedback training (*r* = 0.607, *p* = 0.036) (see Figure 9d). Moreover, the mean strength of connection 2 was significantly increased in the experimental group after the neurofeedback training (see Figure 9f).

## 4. Discussion

In this study, we tested how the upregulation of the right dPMA changed DFNC underlying the ME tasks and how the alteration of DFNC affected motor behavior. We found that the neurofeedback training of the right dPMA changed the temporal properties of two dominant states, which were state 2, with weak network connections, and state 6, with strong MN-MN connections and negative DMN-MN connections. The neurofeedback training induced less occurrence of state 2 and more occurrence of state 6 for the experimental group compared to the control group. For state 2 of the experimental group, the dwell time/fraction time showed negative correlation with the tapping frequency and PSC of the neurofeedback training runs. The alteration of FNC between the hippocampus network and the bilateral precentral network was negatively correlated with the alteration of the finger tapping frequency. The study shows that the upregulation of a specific motor-related region with rtfMRI-neurofeedback is associated with underlying DFNC changes, which are linked to motor improvements.

### 4.1. DFNC Changes after the Neurofeedback Training

In this study, our first goal was to identify DFNC changes in the ME tasks after the neurofeedback training. We found that the upregulation of the right dPMA induced the DFNC changes in state 2 and state 6 that showed the dominant mean dwell time among the seven states in the experimental group. For state 2, the prominent feature was that overall weaker connections existed widely between and within networks. Previous studies found that the dynamic state with weaker connectivity between and within networks was more frequently observed during rest than during tasks [47]. Moreover, the most frequent dynamic FC states during rest showed attenuated connectivity between and within networks [25,26]. The weak FNCs of state 2 may indicate relatively less efficient information transfer between brain networks. Based on the findings of the previous studies, we inferred that dynamic state 2 possibly was relevant to relaxation and rest. For state 6, the connections within MN showed high positive correlation and the connection between MN and DMN showed negative correlation. The high positive correlation within MN and anti-correlation between DMN and MN in state 6 are in line with the previous evidence revealing that the finger tapping task activated the brain regions in the motor-related networks [3,13,48] and DMN was negatively correlated with the other task-positive brain networks [49]. Therefore, we inferred that state 6 may be relevant to the task-focused state.

The mean dwell time and fraction of time of FC states are important temporal properties of dynamic FC. Recent studies have reported significant changes in the mean dwell time in some specific FC states that are associated with aging [28], development [27], training [29], and disorders [30,50]. In contrast to the control group, the experimental group had significantly higher mean dwell time/fraction time in state 6 and significantly lower fraction time in state 2 after the neurofeedback training (see Figure 4). The results indicated that the experimental group tended to spend more time on task-focused state 6 and focused attention on their finger tapping tasks after the neurofeedback training. In contrast, a sham-feedback control group who saw rt-fMRI feedback from another subject, but believed they were receiving feedback from their own right dPMA, tended to spend more time in rest state 2 and were distracted from their finger tapping tasks after the training. The comparison between the two groups suggests that neurofeedback information played an important role in the learning process and changed the temporal variation of FC states of the two groups. The true neurofeedback played a positive role in driving the experimental group more frequently into an attention state, whereas the sham feedback played a negative role in driving the experimental group more frequently into a rest state. Moreover, the results provided converging evidence to support that the temporal properties of DFNC are alterable by training.

### 4.2. Subnetwork Connectivity Changes in the Target FC States after the Neurofeedback Training

For the D-D subnetwork within the DMN of state 2, the experimental group showed significantly lower mean strength than the control group after the training (see Figure 5a). Previous studies demonstrated that DMN showed high activity during mind-wandering [51]. The lower strength of the D-D subnetwork within the DMN of the experimental group suggests that the experimental group focused more attention on the finger tapping tasks than the control group. These results are consistent with the above DFNC findings that the experimental group spent less time in state 2 and more time in state 6.

For the C-D subnetwork between the CCN and DMN, the mean strength of the experimental group was significantly higher than that of the control group in state 2 after the neurofeedback training (see Figure 5b). The prefrontal cortex in the CCN played an important role in execution function [52] and activity in the pre-frontal cortex also could represent goals and the means to achieve them [53]. Based on the role of the CCN, we inferred that the true neurofeedback training possibly facilitated the experimental group to control their own thoughts and actions based on the experimental goals by increasing the interaction between the CCN and DMN, which might lead to the reduction in D-D strength.

### 4.3. The Relation between DFNC and Motor Performance/Training Effect after the Neurofeedback Training

The second goal of this study was to reveal the relationship among DFNC, the neurofeedback training effect, and motor performance. Because the mean dwell time/fraction of time reflects the temporal property of DFNC and the subnetwork strength reflects the network connectivity of DFNC, this study used the mean dwell time/fraction of time and the subnetwork strength to represent DFNC. The training effect was measured as the PSC of each training run and the motor performance was measured as the tapping frequency. For the experimental group, the mean dwell time/fraction of time of state 2 and the D-D strength of state 2 were negatively correlated with the PSC of training runs C and D after the neurofeedback training (see Figure 7 and Figure 8a,b). The higher PSC of training runs C and D caused the subjects in the experimental group to spend less time in state 2 and reduced the D-D strength of state 2 in the post-training ME runs. For the control group with sham feedback, the V-C strength of state 6 was positively correlated with the PCS of training runs C and D after the neurofeedback training, whereas a significant correlation between DFNC of state 2 and the PSC of the training runs was not observed. The different results of the two groups may suggest that the successful upregulation of the PSC in the target ROI in training runs possibly helped the experimental group inhibit the D-D strength and the occurrence of state 2 after the neurofeedback training, which is consistent with the DFNC changes of state 2 induced by the neurofeedback training. For the controls with sham feedback, the subjects with a stronger PSC in training runs C and D possibly had stronger control ability to use visual memory to perform ME tasks, which could lead to high interactions between the control and visual networks in the ME tasks after the training.

The mean dwell time of the experimental group in state 2 was negatively correlated with the tapping frequency after the neurofeedback training (see Figure 6a,b). The longer the experimental subjects remained in state 2, the lower tapping frequency they would have. However, such a correlation was not observed in the experimental group before neurofeedback training or in the control group before and after training. The results indicate that the true neurofeedback training established the relationship between the DFNC of state 2 and motor behavior. Moreover, it should be noted that the relationship between DFNC and the PSC/motor performance was observed, whereas the relationship between the PSC and motor performance was not observed. The results may suggest that the motor improvement after the neurofeedback training could be attributed to the changes in DFNC that were induced by the neurofeedback training.

It should be noted that the temporal properties of state 2 in the experimental group were correlated with both the PSC of the training runs and motor performance after the neurofeedback training, which may indicate that state 2 was an important state that can capture the training effect and motor behavior. The previous reports found that the state that is characterized by sparse links with relatively weak connections and a lack of strong anti-correlations is relevant to development, aging, and disease [27,28,54,55,56]. Our results provided further evidence to support the importance of state 2 with overall weak links. Moreover, it was observed that the frequency of this state was related to the amount of self-focused thought [27]. For the experimental group, the mean dwell time of state 2 during ME tasks was negatively related to the amount of upregulation of the right dPMA, which may suggest that the successful upregulation reduced the time to think about themselves after neurofeedback training. Due to the reduced time for self-focused thought, the experimental group spent more time on task-focused state 6 and improved their finger-tapping frequencies.

### 4.4. Network Connectivity Changes of the Target FC States

For the experimental group, neurofeedback learning mainly increased the connectivities between the MN and VN/DMN and decreased the connectivities between the MN and CCN in state 2 (see Figure 9a). For the control group, the training mainly increased the connectivities between the MN and VN in state 6 (see Figure 9b). Both groups used motor imagery strategies in the training. Because motor imagery and motor execution shared congruence in functional neuroanatomy [57,58], the training mainly resulted in the alteration of connectivity between the MN and the other subnetworks for the two groups. The visual cortex was possibly engaged in forming a visual memory of the motor in the motor imagery training [59]. The control group did not acquire motor skills because they did not find effective strategies to control their brain activities during the neurofeedback training. The increased link between the MN and VN in task state 6 might indicate that the control group possibly needed to make more efforts to use the visual memory of the motor to complete the finger-tapping task after the training. In contrast, the experimental group was able to find efficient strategies and acquired motor skills during the neurofeedback training. They did not need to make great efforts to use visual memory to perform finger tapping due to their learned motor skills during the training. Thus, the connectivities between the MN and VN in task state 6 were not observed in the experimental group. Moreover, the neurofeedback training caused the experimental group to increase the connectivities between the MN and VN in rest state 2. The increase in one connectivity (connection 2) between the MN and VN was positively correlated with the increase in the tapping frequency for the experimental group. The results might indicate that the true neurofeedback caused the experimental group to form implicit visual memory of the motor behavior. The experimental group possibly tended to use visual implicit memory in the relaxation state (state 2), which could have contributed to the improvement in motor behavior.

For the connection between the bilateral precentral gyrus component and the hippocampus component (connection 1) in state 2, neurofeedback training led to a significant decrease in the experimental group (see Figure 9e). The connectivity of connection 1 in the experiment changed from positive to negative after the neurofeedback training. A larger decrease in the connectivity of connection 1 led to a higher increase in the tapping frequency after the neurofeedback training of the experimental group. Before the training, all subjects needed to remember the sequence order of the fingers when they performed the finger-tapping task. Because the hippocampus gyrus plays an important role in the formation of new memories [60] and the bilateral precentral gyrus is involved in executing voluntary motor movements [16], the hippocampus component showed positive correlation with the precentral component to form the memory of the finger sequence and contribute to the finger tapping before training. After the neurofeedback training, the correlation between the hippocampus component and the precentral component was significantly decreased and became negative for the experimental group, whereas the correlation was still positive for the control group. The anti-correlation between the hippocampus component and the precentral component after neurofeedback training may suggest that the precentral component of the experimental group showed competition with the hippocampus component. Because the experimental subjects acquired the tapping skill during neurofeedback training, they might have used implicit visual memory and executed the finger tapping more automatically after the training, which possibly resulted in the competition between the precentral and hippocampus components.

### 4.5. Limitations

There are some caveats in this study. First, this study focused on investigating the two core dynamic states because the two states are dominant in the fluctuations of FNC. Future studies should examine more dynamic states to better understand the DFNC underlying the ME task. Second, the number of participants in this study was not large, which may lead to the instability of the results to some extent. In order to guarantee the reliability of the results, we reported the significant results with large effect size. Moreover, it is necessary to collect more subjects to investigate the alteration of DFNC induced by the neurofeedback training in future studies.

## 5. Conclusions

This study investigated the alteration of DFNC underlying ME tasks after neurofeedback training of the right PMA. Findings show that the upregulation of the right PMA induced DFNC changes such as reduced mean dwell time in the weak-connected state 2 and an increased fraction of time in state 6 with strong MN-MN connections. Moreover, the mean dwell/fraction of state 2 was associated with both the amount of upregulation and the improvements in motor performance. Thus, the findings may suggest that the rtfMRI neurofeedback of a key motor-related region can improve the motor performance by changing the temporal properties of DFNC underlying ME tasks.

## Figures and Tables

**Figure 1 brainsci-11-00582-f001:**
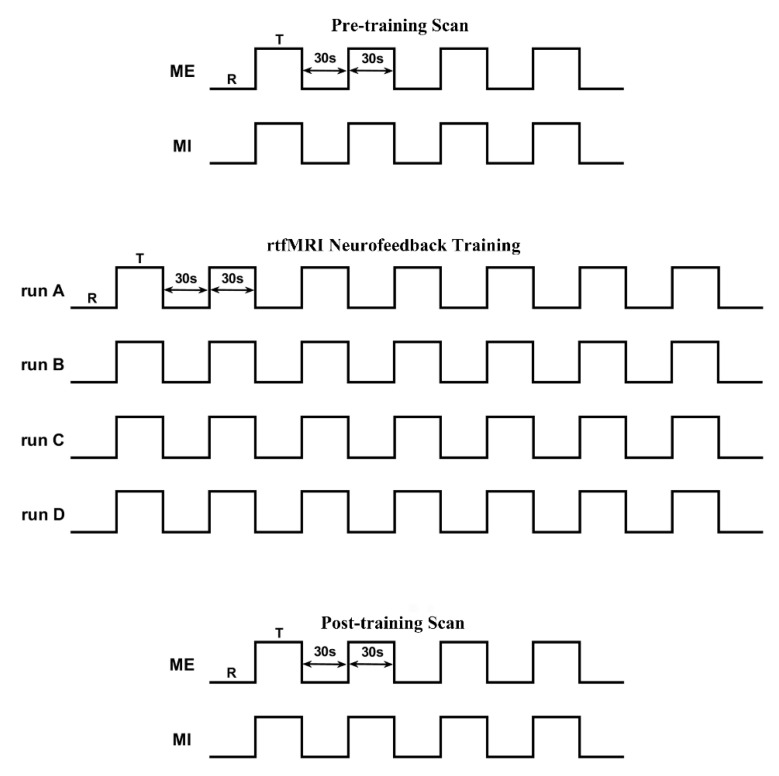
The diagram of the fMRI experiment.

**Figure 2 brainsci-11-00582-f002:**
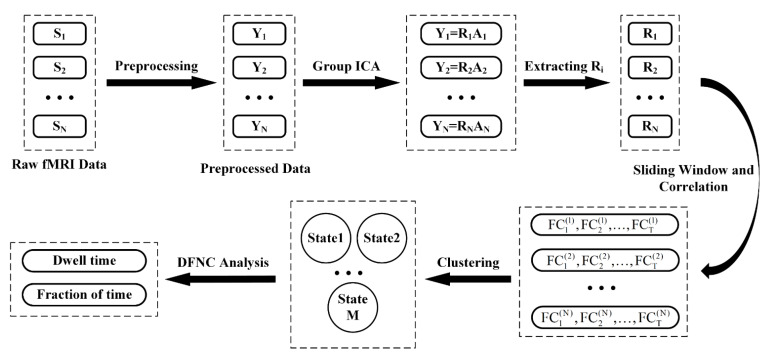
The overview of the framework of DFNC processing. S_i_ represents the raw fMRI data of the i^th^ subject. Y_i_ represents the preprocessed fMRI data of the i^th^ subject. R_i_ represents the mixing matrix (time courses) and A_i_ represents the source matrix (independent components) of the i^th^ subject. FC_j_^(i)^ represents the FC pattern of the j^th^ sliding window of the i^th^ subject.

**Figure 3 brainsci-11-00582-f003:**
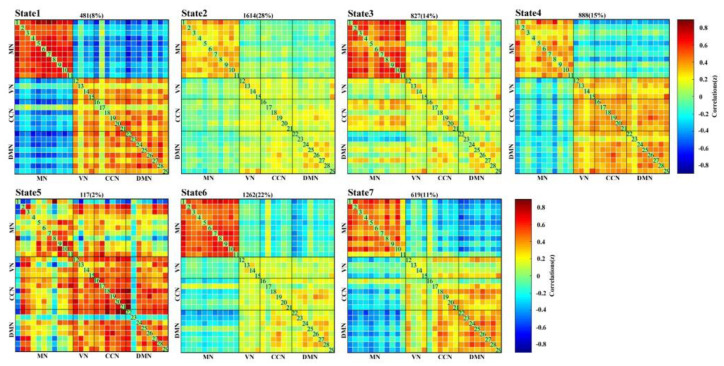
The median pattern of 7 FC states in motor execution tasks run clustered by K-means. The total number and percentage of occurrences is listed above each FC pattern.

**Figure 4 brainsci-11-00582-f004:**
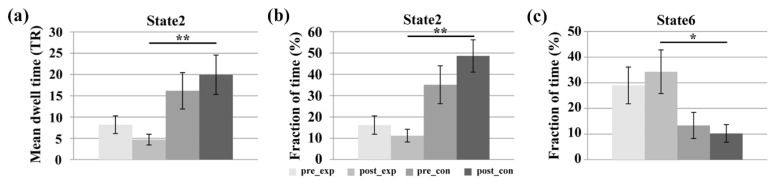
Time measurements for the pre-training and post-training ME runs of the two groups. (**a**) The mean dwell time of state 2, (**b**) the fraction of time of state 2, and (**c**) the fraction of time of state 6. Pre_exp represents the pre-training run of the experimental group. Post_exp represents the post-training run of the experimental group. Pre_con represents the pre-training run of the control group. Post_con represents the post-training run of the control group. Error bars represent the standard error (* *p* < 0.05; ** *p* < 0.01).

**Figure 5 brainsci-11-00582-f005:**
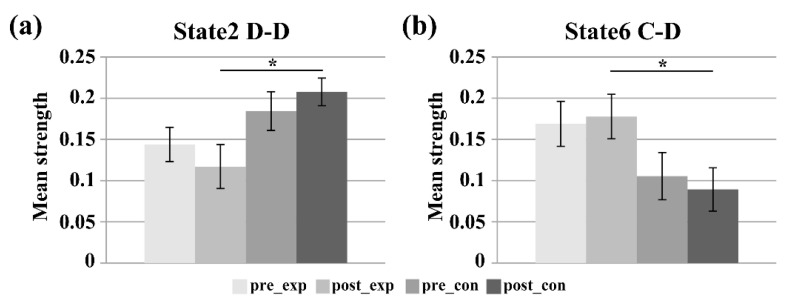
The mean strength of the sub-networks showing significant group/training differences. (**a**) The D-D subnetwork in state 2, and (**b**) the C-D subnetwork in state 6. Error bars represent the standard error (* *p* < 0.05).

**Figure 6 brainsci-11-00582-f006:**
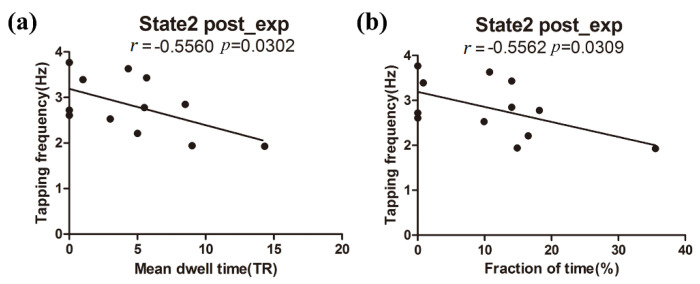
The relationship between tapping frequency and DFNC for post-training ME run of the experimental group. (**a**) The significant regression relationship between the tapping frequency and the mean dwell time of state 2, and (**b**) the significant regression relationship between the tapping frequency and the fraction time of state 2.

**Figure 7 brainsci-11-00582-f007:**
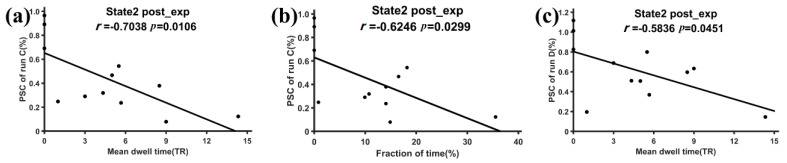
The relationship between the PSC of the neurofeedback runs and DFNC of the post-training run for the experimental group. (**a**) The significant regression relationship between the mean dwell time of state 2 and PSC of run C, (**b**) the significant regression relationship between the fraction time of state 2 and PSC of run C, and (**c**) the significant regression relationship between the mean dwell time of state 2 and PSC of run D.

**Figure 8 brainsci-11-00582-f008:**
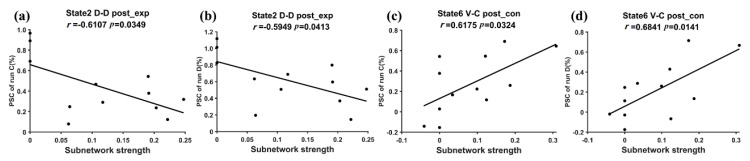
The relationship between the PSC of the neurofeedback runs and the subnetwork strength of the post-training run. (**a**) The significant regression relationship between the PSC of run C and the strength of D-D subnetwork for the experimental group’s state 2. (**b**) The significant regression relationship between the PSC of run D and the strength of D-D subnetwork for the experimental group’s state 2. (**c**) The significant regression relationship between the PSC of run C and the strength of the V-C subnetwork for the control group’s state 6. (**d**) The significant regression relationship between the PSC of run D and the strength of the V-C subnetwork for the control group’s state 6.

**Figure 9 brainsci-11-00582-f009:**
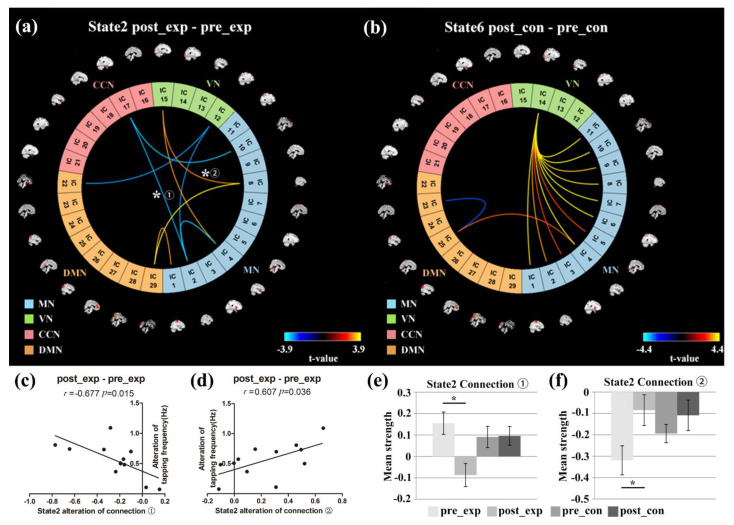
Significant connectivity differences in the target FC states for the experimental group and control group. (**a**,**b**) The connectives showing significant differences for the post-training minus the pre-training run in state 2 of the experimental group (**a**) and in state 6 of the control group (**b**), (**c**) the significant regression relationship between the alteration of the tapping frequency and the alteration of the connectivity of connection 1 for the experimental group after neurofeedback training, (**d**) the significant regression relationship between the alteration of the tapping frequency and the alteration of the connectivity of connection 2 for the experimental group after neurofeedback training, and (**e**,**f**) the mean connectivity strength of connection 1 (**e**) and connection 2 (**f**) for the pre-training and post-training runs of the two groups. Error bars represent the standard error (* *p* < 0.05).

**Table 1 brainsci-11-00582-t001:** Statistics and specific brain locations of connections that survived the NBS approach in state 2 of the experimental group for the post-training run versus the pre-training run.

Component Index 1	Network 1	Region 1	Component Index 2	Network 2	Region 2	*t*-Score	*p*-Value
1	MN	Left postcentral gyrus	29	DMN	Cuneus	3.227	0.0081
2	MN	Bilateral precentral gyrus	4	MN	Bilateral inferior frontal gyrus, triangular part	−3.349	0.0065
2	MN	Bilateral precentral gyrus	12	VN	Bilateral middle occipital gyrus, superior occipital gyrus	−3.126	0.0096
2	MN	Bilateral precentral gyrus	17	CCN	Hippocampus	−3.276	0.0074
4	MN	Bilateral inferior frontal gyrus, triangular part	15	VN	Bilateral middle occipital gyrus	3.268	0.0075
8	MN	Bilateral postcentral gyrus	15	VN	Bilateral middle occipital gyrus	3.109	0.0099
8	MN	Bilateral postcentral gyrus	29	DMN	Cuneus	3.934	0.0023
10	MN	Supplementary motor area	17	CCN	Hippocampus	−3.381	0.0061
12	VN	Bilateral middle occipital gyrus, superior occipital gyrus	22	VN	Cingulate and paracingulate gyrus	−3.142	0.0094

**Table 2 brainsci-11-00582-t002:** Statistics and specific brain locations of connections that survived the NBS approach in state 6 of the control group for the post-training run versus the pre-training run.

Component Index 1	Network 1	Region 1	Component Index 2	Network 2	Region 2	*t*-Score	*p*-Value
1	MN	Left postcentral gyrus	29	DMN	Cuneus	3.227	0.0081
2	MN	Bilateral precentral gyrus	4	MN	Bilateral inferior frontal gyrus, triangular part	−3.349	0.0065
2	MN	Bilateral precentral gyrus	12	VN	Bilateral middle occipital gyrus, superior occipital gyrus	−3.126	0.0096
2	MN	Bilateral precentral gyrus	17	CCN	Hippocampus	−3.276	0.0074
4	MN	Bilateral inferior frontal gyrus, triangular part	15	VN	Bilateral middle occipital gyrus	3.268	0.0075
8	MN	Bilateral postcentral gyrus	15	VN	Bilateral middle occipital gyrus	3.109	0.0099
8	MN	Bilateral postcentral gyrus	29	DMN	Cuneus	3.934	0.0023
10	MN	Supplementary motor area	17	CCN	Hippocampus	−3.381	0.0061
12	VN	Bilateral middle occipital gyrus, superior occipital gyrus	22	VN	Cingulate and paracingulate gyrus	−3.142	0.0094
1	MN	Left postcentral gyrus	29	DMN	Cuneus	3.227	0.0081
2	MN	Bilateral precentral gyrus	4	MN	Bilateral inferior frontal gyrus, triangular part	−3.349	0.0065
2	MN	Bilateral precentral gyrus	12	VN	Bilateral middle occipital gyrus, superior occipital gyrus	−3.126	0.0096
2	MN	Bilateral precentral gyrus	17	CCN	Hippocampus	−3.276	0.0074

## Data Availability

The datasets generated for this study are available on request from the corresponding author.

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
