# Peer review of "Dynamic Functional Network Connectivity Changes Associated with fMRI Neurofeedback of Right Premotor Cortex"

_brainsci, 2021, doi:10.3390/brainsci11050582_

Round 1

Reviewer 1 Report

The manuscript titled: ‘Dynamic functional network connectivity changes associated with fMRI neurofeedback of right premotor cortex’ by Zhiying Long et al., is an interesting study concerning possibility  of regulation of target network connectivity to improve behavioural performance. I have some questions.

‘Materials and Methods’ section

Please provide information, why right-handed people were chosen?

Page 3, line 105, whether the other participants were women?

‘Discussion’ section

Page 15 , line 566, page 16 line 586, please explain, which brain connections may be related to ‘unconsciously’ ,

Page 13, line 474, page 15,  line 534, please provide information, which disorders it concerns,

I am asking authors for comment, whether rtfMRI neurofeedback would be helpful in anxiety disorders (for example post- traumatic stress disorders, PTSD) therapy?

Reviewer 2 Report

This is an important manuscript demonstrating the effect of fMRI neurofeedback on functional network connectivity. Here are a few suggestions to help improve the manuscript.

1)There are several successful denoising fMRI procedures. Did the authors use any denoising of the fMRI data or removal of physiological noise?

2)For this sentence in the Discussion section

“For the state 2 of the experimental group, the dwell time/fraction time showed negative correlation with the tapping frequency and PSC of the neurofeedback training runs.” The authors need to expand this statement a lot to explain how dwell time, tapping frequency, and PSC relate to the original goal to measure the relationship between network connectivity, training effect and motor performance. The way this reads right now, its way too vague and obtuse.

3) For the relationship between DFNC, training effect and motor performance, the authors could do a direct correlation between DFNC subnetwork strength and training effect and DFC subnetwork strength and motor performance.
